# Decouped Variational Graph Autoencoder for Link Prediction

## ABSTRACT

Link prediction is an important learning task for graph-structured data, and has become increasingly popular due to its wide application areas. Graph Neural Network (GNN)-based approaches including Variational Graph Autoencoder (VGAE) have achieved promising performance on link prediction outperforming conventional models which use hand-crafted features. VGAE learns latent node representations and predicts links based on the similarities between nodes. While the inner product based decoder effectively utilizes the node representations for link prediction, it exhibits sub-optimal performance due to the intrinsic limitation of the inner product. We found that the the cosine similarity and norm simultaneously try to explain the link probability, which hinders the gradient flow during training. We also point out the message passing scheme is unexpectedly dominated by the nodes with large norm values. In this paper, we propose a stochastic VGAE-based method that can effectively decouple the norm and angle in the embeddings. Specifically, we relate the cosine similarity and norm to two fundamental principles in graph: *homophily* and *node popularity* respectively. Following the principles in graph, we define a generative process in the VGAE framework. Our learning scheme is based on a hard expectation maximization learning method; we infer which of the two has been exerted for link formation, and subsequently optimize based on this guess. We comprehensively evaluate our proposed method on link prediction task. Through extensive experiments on real-world datasets, we demonstrate our model outperforms the existing state-of-the-art methods on link prediction and achieves comparable performances on other downstream tasks such as node classification and clustering. Our code is at https://anonymous.4open.science/r/dvgae-A0B4.

## CCS CONCEPTS

• **Computing methodologies** → **Neural networks**; *Learning latent representations*;

## KEYWORDS

Link Prediction, Graph Neural Networks, Variational Graph Autoencoder

**ACM Reference Format:**
Anonymous Author(s). 2018. Decouped Variational Graph Autoencoder for Link Prediction. In *Proceedings of Make sure to enter the correct conference title from your rights confirmation emai (WWW '24)*. ACM, New York, NY, USA, 11 pages. https://doi.org/XXXXXXX.XXXXXXX

## 1 INTRODUCTION

Graph-structured data is omnipresent in various fields, such as citation networks, social networks, recommender systems, and knowledge graphs. In graph-structured data, links reflect the relation between the nodes, where nodes in such applications can be documents, web users, items, or concepts. One of the main challenges with graph-structured data is link incompleteness, where many edges are missing or unobserved. Due to this nature, link prediction is one of the critical tasks in network analysis and has attracted increasing attention. Under the homophily assumption [10, 34, 52], link prediction approaches attempt to estimate the link through evaluating the similarity within a pair of nodes based on observed links and the associated attributes of nodes.

The link prediction problem has been a long-standing challenge and has been extensively studied within academia and industry. The recent success of Graph Neural Networks (GNNs) has boosted research on various graph learning tasks, including capturing the relations between nodes in graph-structured data. Several recent studies present promising results on link prediction [5, 21, 22, 39, 48, 54]. Methods adopting GNNs, including Graph Convolutional Network (GCN) [23] automatically learn latent representations from node attributes and their local neighborhoods. The core of GNN lies in the message-passing scheme [21, 48], where the the node embeddings are passed along the edges of the graph. GNN-based approaches replace hand-crafted algorithms to optimize graph and permits more flexible modeling. Variational Graph Autoencoder (VGAE) [22] is a variational probabilisitc generative framework with GCN. With its flexibility and proven performance, various extensions have been proposed within this framework. However, they exhibit underperformance in link prediction on low-degree and high-degree nodes. We hypothesize that this is due to the intrinsic limitation of inner-product decoder in VGAE.

The inner-product can be decomposed into the cosine similarity (or normalized inner product) and norm, where these two components compete each other simultaneously trying to explain the link probability. This becomes problematic when one of the components becomes easier to perform gradient descent during backpropagation, which motivates our study. Figure 1a shows the norm of node embeddings from Cora and PubMed datasets obtained via VGAE with respect to the node degrees. For the nodes with high degree or low degree, the loss with inner-product can be easily compensated through the norm values. In Figure 1a, we observe low-degree nodes tend to have small norms in their node embeddings, and vice versa. For high-degree nodes, the norm of node embedding gets high to account for the high number of edges; thus cosine similarity has minimal effect on the link probability. The same logic applies to the opposite case with low-degree nodes. For high-degree and low-degree nodes, optimizing the node embeddings in terms of the direction in vector space can be challenging. However, the high norm (magnitude) value causes a negative impact on message passing scheme, where messages are aggregated from the neighboring nodes. Specifically, we illustrate how high norm of node embedding

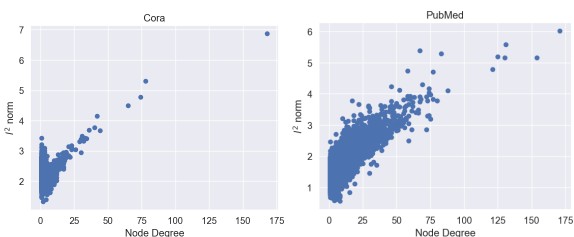

(a) $l_2$ **norm of z in Cora (left) and PubMed (right)**

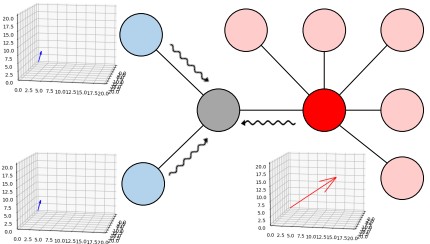

(b) **Messages are aggregated at the target vertex (gray). The vector in 3-d box represents the node embedding vector obtained through VGAE; the color (red/blue) represents the node attribute.**

Figure 1: (a) Degree of nodes and their $\|z\|$s learned from VGAE. The norm of node embedding increases respect to its degree. (b): Node embedding with large norm unexpectedly dominates the message passing to the node in gray.

harms the message passing in Figure 1b. When the messages are propagated to the target node in gray, the node with high norm unexpectedly dominates the messages. As such, when the high-degree node happen to behave differently with embedding vector heading toward different direction, the message passing scheme can hinder the learning of node embeddings.

To solve the aforementioned problems, we propose a novel generative algorithm to decouple the two components in the inner product within VGAE framework. Specifically, we incorporate two different embedding spaces, namely the embedding space for *homophily* with the cosine similarity based decoder and the embedding space for *node popularity* with the norm based decoder. Through this approach, we can effectively learn each embedding and account for the links independently. Moreover, when decoupled, we can also restrict the message passing only on angular embeddings to avoid the *domination* effect in Figure 1b. However, even with the separate-space approach, decoupling the two effects is not trivial as the two effects are not observable. A simple remedy is to focus only on one (homophily) by ignoring the other (node popularity). While it is effective than vanilla-VGAE, it is still sub-optimal. Our model considers both components individually through decoupling the two properties through the proposed stochastic generative process. We also propose a hard Expectation-Maximization (EM) algorithm to perform end-to-end learning. Our model achieves state-of-the-art (SOTA) results in link prediction on attributed networks.

We summarize the main contributions of our work as follows:

- We discuss the intrinsic limiation of VGAE decoder, and propose an end-to-end approach within the framework of VGAE, which decouples norm and angular node embeddings through the proposed generative process.
- We comprehensively evaluate our method on numerical experiments and show that it consistently outperforms the existing state-of-the-art link prediction models.
- We additionally show how the latent embeddings learned only from the normalized form excluding the norm component performs for link prediction, which already achieves better results than the current SOTA model.

## 2 RELATED WORK

*Link prediction.* While the work in link prediction spans many fields over a long period of time, we review three major streams of research in link prediction: *heuristic methods, network embedding methods,* and *graph neural networks.* Heuristic methods compute the likelihood of an edge to appear based on different heuristic metrics between nodes within a pair. Common neighbor [29], Katz index [19], and PageRank [4] rely on hand-crafted rules. Network embedding methods map nodes in a network to lower dimensional spaces while effectively preserving the network structure. Random-Walk based models such as DeepWalk [40] and Node2Vec [13] define notations of the node's neighborhood and learn latent space representations of social interactions. Methods adopting GCN automatically learn node embeddings for link prediction and other downstream tasks, such as node classification and community detection. Our model and the baseline approaches in this study perform link predictions using graph neural networks. We later review the link predictions with GNNs as baseline models in more detail (see Section 5.2).

*Node Popularity.* Node popularity has been frequently discussed in the community detection literature prior to the introduction of GNNs. In [18], the authors proposed the degree-corrected blockmodel for community detection. In [24], heterogeneity of actor degrees has been taken into account for latent cluster random effects models. The authors in [12] integrated node popularity into the mixed membership stochastic blockmodels [2], which is perhaps the closest to ours in spirit. In their approach, the probability of a link is defined by the membership assignment and the node popularity through linear summation. Our approach however decouples the two effects and lets one of each effect attend to the probability of a link, which avoids the smoothing of the two. The decoupling can also prevent message passing the high-magnitude embeddings. Node popularity has been more frequently accounted for in the Recommender System (RS) literature. Previous studies [6, 27, 30, 46] in this literature incorporated item popularity to RS. To the best of our knowledge, this is the first in VGAE to incorporate node popularity. We emphasize that our proposed model is not just an extension of previous studies in homophily and node popularity, but rather a study finding the problem of the competition between norm and angular embeddings. We associate them with two popular phenomena in network science simply for better explanation.

 

# 3 PRELIMINARIES

## 3.1 Problem Formulation

We are interested in the problem setting where we are given an undirected, unweighted graph $\mathcal{G} = (\mathcal{V}, \mathcal{E})$ with $N = |\mathcal{V}|$ nodes and a $N \times D$ feature matrix $\mathbf{X}$. Let $\mathbf{A}$ be an adjacency matrix of $\mathcal{G}$, and $\mathbf{D}$ be its degree matrix. With these settings, the goal is to predict whether an edge exists within an unobserved pair of nodes based on $\mathbf{X}$ and observed links with $\mathbf{A}$. This can be achieved by learning the $F$-dimensional node embeddings (or stochastic latent variables) $\mathbf{z}_i$ that best reconstruct the network $\hat{\mathbf{A}}$, where we can summarize the $\{\mathbf{z}_i\}$ in an $N \times F$ matrix $\mathbf{Z}$.

## 3.2 Background: VGAE For Link Prediction

*Variational Graph Autoencoder.* VGAE [22] tries to solve the problem of observed link predictions by learning the node embeddings (or latent variables) $\mathbf{z}_i$ that best reconstruct the network $\hat{\mathbf{A}}$. VGAE learns a distribution over the latent space for each input, where $\mathbf{z}_i$ is sampled from the distribution. Given $\mathbf{Z} = \{\mathbf{z}_1, \ldots, \mathbf{z}_N\}$, a simple inner-product decoder in VGAE reconstructs the adjacency matrix as $\hat{\mathbf{A}} = \sigma(\mathbf{Z}\mathbf{Z}^\top)$ with a Gaussian prior. Thus, the encoder $q_\phi(\mathbf{Z} \mid \mathbf{X}, \mathbf{A})$ becomes the inference model which learns the variational posterior; the decoder $p_\theta(\mathbf{A} \mid \mathbf{Z})$ becomes the generative model.

*Inference model.* Due to the intractability of the marginal likelihood, true posterior is approximated by a Gaussian distribution in variational inference [22, 50].

$$q_\phi(\mathbf{z}_i \mid \mathbf{X}, \mathbf{A}) = \mathcal{N}(\boldsymbol{\mu}_i, \mathrm{diag}(\boldsymbol{\sigma}_i^2)), \qquad (1)$$

where the mean vector and variance vector for Gaussian distribution are parameterized by a two-layer graph convolutional network (GCN) [23]: $\boldsymbol{\mu}_i = \mathrm{GCN}_\mu(\mathbf{X}, \mathbf{A})$ and $\log \boldsymbol{\sigma}_i = \mathrm{GCN}_\sigma(\mathbf{X}, \mathbf{A})$. Here, the node features are naturally incorporated thorough input $\mathbf{X}$.

*Generative Model.* In the generative model, the similarity are computed across all pairs of nodes, where the inner product is used along with sigmodal function.

$$p(A_{ij} = 1 \mid \mathbf{z}_i, \mathbf{z}_j) = \sigma(\mathbf{z}_i^\top \mathbf{z}_j), \qquad (2)$$

where the probability of a link between node $i$ and $j$ is determined by the similarity between the node embeddings. Here, the inner product is used for similarity measure, which is fed into sigmoid.

*Observation.* The inner-product in the decoder is a simple yet effective method, where the VGAE and its extensions have been achieving SOTA performances. However, we found that the inner-prodcut decoder likely results in sub-optimal performance when performed on low-degree nodes or high-degree nodes. We hypothesize that the two factors, namely norm and cosine similarity, provide different and complementary effects to the inner product while the two effects are tightly coupled and trained jointly. This observation and the property motivate us to decouple the two effects for better training. In this study, we correspond the norm to *node popularity*; the cosine similarity to *homophily*. Thus, the main research question we consider is **RQ1)** how to decouple the node popularity and homophily in graph structured data. Decoupling the two property is not trivial as the two simultaneously try to account for the probability of link generation. In backpropagation, the gradient can still

flow to any direction unless specified. This becomes problematic when one of the factor is favored dominantly in backpropagation over the other factor. Similar observation also has been discussed in [1], where only the norms of node embeddings gets close to zero for nodes with small degree. Their remedy to this problem was using the normalized embeddings (or the cosine similarity). However, the approach in [1] can only account for the *homophily*. Their experimental results also reveal how the normalized embeddings become effective especially when the network is sparse. While it effectively addresses the problems with near-zero-degree nodes, due to the additional constraints, it becomes less flexible than the vanilla-VGAE. In fact, the model in [1] exhibits performance degradation on link predictions associated with high-degree nodes (see Appendix B.1).

When the node popularity and homophily can be decoupled, it brings us to the second research question. We are interested to see **RQ2)** whether the embedding for homophily itself becomes more accurate when the node popularity effect gets removed during training. When the two effects coexist and are trained jointly, we suspect that the two effects interfere each other, which may distort the node embeddings. As such, we expect to have more accurate embedding when the node popularity gets removed during training. However, this is not trivial as the property (node popularity vs homophily) that contributes to generation of link is not observable. To this end, we propose a novel approach that can decouple the node popularity and the homophily for link prediction. While there have been some studies in recommendation system literature using GNN for addressing item popularity [8, 28, 30], to the best of our knowledge, this is the first study to incorporate node popularity in VGAE for link prediction.

# 4 PROPOSED METHOD

Here, we address the aforementioned questions by introducing a model within VGAE framework. Our main idea is threefold; (a) we consider two properties in graph namely *node popularity* and *homophily* for link predictions, (b) we propose an EM-like learning algorithm that alternates between estimating the associated effects from the two and learning their embeddings, (c) we stochastically estimate the respective effect and achieve more attended representations.

The term *node popularity* also appears in other literature, and we introduce our definition of node popularity for this study.

**Definition 1** (Node Popularity in Graph). For a graph $\mathcal{G} = (\mathcal{V}, \mathcal{E})$, given two nodes $i, j \in \mathcal{V}$, a link can be generated even when there is **no similarity** between node $i$ and $j$. An undirected link is defined by a node popularity function of node $i$ or $j$.

In our generative process, at each interaction between a given pair, one of the *promising* scenario under homophily or node popularity is stochastically sampled. This is particularly a natural idea, where we observe frequently in real-world. In social network, users also make friends with or follow *popular* users without sharing common interest. In this section, we elaborate our generative model in VGAE framework, and provide the learning algorithm. Our model decouples the node embeddings into two components: norm and angular, which corresponds to the decoder only using the norms and the decoder only using the cosine similarities separately. We name

our model "Decoupled VGAE (D-VGAE)". The overall framework and its learning process is illustrated in Figure 2.

## 4.1 Generative Process

*D-VGAE generative process.* We assume each node can establish a link with others under one of the two phenomena: homophily or node popularity. For each pair, instead of stochastically selecting one of the two phenomena directly, we first sample the value (in binary) of interaction through Bernoulli with respect to the similarity measure between the two nodes. We take this approach for three reasons: (1) two phenomena are not observable and cannot be compared directly unless one of the function is pre-given; (2) homophily can be directly inferred from the node features, while the node popularity can be inferred indirectly by removing homophily; and (3) *decoder collapse* (two decoders behave similarly) can be better prevented. We start by extending the embedding vector $\mathbf{z}$ from VGAE to $\mathbf{z}^p$ and $\mathbf{z}^h$ for node popularity and homophily respectively, and further define the generative process. The overall generative process can be summarized as below:

- For each node $i \in \mathcal{V}$, sample node latent variables for homophily: $\mathbf{z}_i^h \sim \mathcal{N}(\boldsymbol{\mu}^h, \Sigma^h)$ .
- For each node $i \in \mathcal{V}$, sample node latent variables for node popularity: $\mathbf{z}_i^p \sim \mathcal{N}(\boldsymbol{\mu}^p, \Sigma^p)$ .
- For each pair of node $i$ and $j$, draw a binary undirected link from a Bernoulli distribution through two-stage process:
  (1) **Homophily**: Draw a binary undirected link from a Bernoulli

$$A_{ij} \sim \text{Bernoulli}(\sigma(\text{sim}(\mathbf{z}_i^h, \mathbf{z}_j^h))), \quad (3)$$

  where $\sigma(\cdot)$ is a sigmoid function, and $\text{sim}(\cdot)$ is a similarity measure function.
  (2) **Node Popularity**: If no link were sampled in stage 1, draw a binary undirected link

$$A_{ij} \sim \text{Bernoulli}(\sigma(\text{snp}(\mathbf{z}_i^p, \mathbf{z}_j^p))), \quad (4)$$

  where $\text{snp}(\cdot)$ is a function that measures the strenght of node popularity.

In vanilla-VGAE, we identify that the engagement of norm and cosine similarity within the inner product hinders the learning of node embedding. Through our generative process, we expect the homophily ($\text{sim}(\cdot)$) only account for the quasi-cosine similarity; and expect the node popularity ($\text{snp}(\cdot)$) only account for the quasi-norm. Thereby, we use normalized inner product for $\text{sim}(\cdot)$ and summation of each of the strength of given pair for $\text{snp}(\cdot)$. More details on $\text{snp}(\cdot)$ is provided in the Appendix B.2. Through the two-stage generative process we propose, homophily and node popularity can be decoupled and considered separately. In the following, we elaborate the above through inference model and generative model within the VGAE framework. To fully verify our hypothesis, we add only necessary changes incorporating our scheme. Finding better posterior distribution can be found in [16], and also in [20, 38, 41] from more general VAE literature. We also refer the readers to [14, 16, 43, 45, 47] for designing better decoder than a simple inner product.

*Inference model.* We use a probabilistic encoder to perform variational posterior inference. The posterior distribution (Eq. 1) is parameterized by encoder which generates the mean $\boldsymbol{\mu}_i$ and the log $\boldsymbol{\sigma}_i$. As our proposed generative process requires $\mathbf{z}_i^h$ and $\mathbf{z}_i^p$, we let our probabilistic encoder generate two sets of variational parameters in the same sense. For learning the embedding of $\mathbf{z}^p$, we remove the message passing scheme, as node popularity is the characteristic of a node itself and to prevent the *domination* effect. We use the encoder from VGNAE [1] as one of the building blocks in our implementation. The $l^2$-normalization in VGNAE can naturally account for the cosine similarity (or normalized inner-product) assumption for homophily. To accomplish this $l^2$-normalization, a GCN called *graph normalized convolutional network* (GNCN) was proposed in [1], which is defined as follows:

$$\text{GNCN}(\mathbf{X}, \mathbf{A}, s) = s\mathbf{D}^{-\frac{1}{2}}\mathbf{A}\mathbf{D}^{-\frac{1}{2}}g(\mathbf{X}\mathbf{W}), \quad (5)$$

where $g([\mathbf{h}_1, \mathbf{h}_2, ..., \mathbf{h}_n]^\top) = [\frac{\mathbf{h}_1}{\|\mathbf{h}_1\|}, \frac{\mathbf{h}_2}{\|\mathbf{h}_2\|}, ..., \frac{\mathbf{h}_n}{\|\mathbf{h}_n\|}]^\top$, $\mathbf{W}$ is a learnable weight matrix, and $s$ is a scaling factor which is set to 1.8 in the following experiments. In a nutshell, normalization in the encoder of GNAE brings same effect as having cosine similarity based decoder. Using GNCN as our encoder can also verify the effectiveness of our proposed scheme over the current SOTA model: VGNAE.

*Generative model.* Through reparameterization trick [22], $\mathbf{z}^p$ and $\mathbf{z}^h$ are sampled from the variational distribution. Given the $\mathbf{z}^p$ and $\mathbf{z}^h$ for all the nodes, every possible pair are compared through two-stage process. We first draw a binary undirected link using the Gumbel trick [17], where we have $p(A_{ij} = 1) = \sigma(\mathbf{z}_i^{h\top}\mathbf{z}_j^h)$. When a positive link is sampled at this stage, it becomes the final link for $\hat{\mathbf{A}}$; otherwise a binary undirected link can be sampled with $p(A_{ij} = 1) = \sigma((\mathbf{z}_i^p + \mathbf{z}_j^p)^\top[1, 0])$, where $\{\mathbf{z}_i^p\}$ in an $N \times 2$ matrix $\mathbf{Z}^p$. We elaborate why we use two-component vector $\mathbf{z}^p$ and take one scalar for computing strength of node popularity treating the other as dummy. The encoder generates normalized vectors $\mathbf{z}^h$ through GNCN. When performing inner product with normalized vectors, the inner product in sigmoid function becomes bounded within acceptable limits through scaling factor. The value $\mathbf{z}^p$ should be bounded as such, otherwise model loses its stability. Therefore, we use the two-component vector for $\mathbf{z}^p$ for normalization, and only use one of the component: $\mathbf{z}^{p\top}[1, 0]$. We can naturally have a bounded values of $\mathbf{z}^p$ by separately normalizing the given vector through GNCN. The links of interest in the literature is the undirected link, and we can sum the two node properties to account for the link due to node popularity. We also only perform message passing on $\mathbf{z}^h$ but not on $\mathbf{z}^p$. This is because node polularity is not a feature that propagates through its neighbor, but rather an unique value of a given node itself. Most of all, this avoids the domination effect from Figure 1b. Our additional experimental results in Appendix B.3 justifies our idea. The most closest to ours is [12], which integrates node popularity with community detection. However, their approach tries to incorporate node popularity in a mixed manner, while we are more interested in decoupling the two effects. We also incorporate their idea into VGAE framework for further analysis and discuss on the differences in Section 5.4.

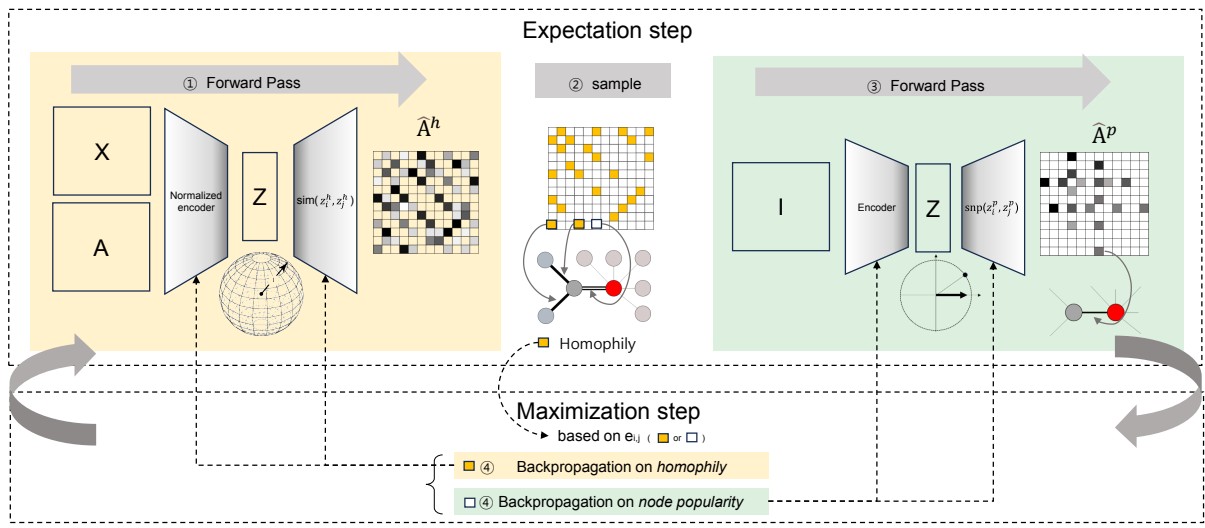

**Figure 2: The overview of D-VGAE's learning scheme.**

## 4.2 The Variational Bound

*Evidence Lower-Bound.* For the inference and learning, we optimize the evidence lower bound (ELBO) with respect to the variational parameters $\boldsymbol{\phi}$.

$$
\begin{aligned}
\log p(\mathbf{A}) \geq \; & \mathbb{E}_{q_\phi(\mathbf{Z}^{p,h}|\mathbf{X},\mathbf{A})}[\log p_\theta(\mathbf{A} \mid \mathbf{Z}^{(p,h)})] \\
& - \text{KL}(q_\phi(\mathbf{Z}^p \mid \mathbf{I},\mathbf{A})||p(\mathbf{Z}^p)) - \text{KL}(q_\phi(\mathbf{Z}^h \mid \mathbf{X},\mathbf{A})||p(\mathbf{Z}^h)) \\
& \stackrel{\text{def}}{=} \mathcal{L}(\boldsymbol{\phi},\boldsymbol{\theta};\mathbf{A}) \, .
\end{aligned}
\tag{6}
$$

The first term of RHS in Equation 6 can be reformulated into following expression:

$$
\mathbb{E}_{q_\phi(\mathbf{Z}^{p,h}|\mathbf{X},\mathbf{A})} \log p_{\theta^h}(\mathbf{A} \mid \mathbf{Z}^h) p^h + \mathbb{E}_{q_\phi(\mathbf{Z}^{p,h}|\mathbf{X},\mathbf{A})} \log p_{\theta^p}(\mathbf{A} \mid \mathbf{Z}^p) q^h,
\tag{7}
$$

where $p_{\theta^h}(\mathbf{A} \mid \mathbf{Z}^h)$ and $p_{\theta^p}(\mathbf{A} \mid \mathbf{Z}^p)$ follow Equation 3 and 4 respectively, which accounts for homophily and node popularity. The $p^h$ and $q^h = 1 - p^h$ are Bernoulli parameters which reflects the probability of a link generated under homophily. We rely on EM-like learning algorithm to estimate each probability, where we provide the details in the following section. The second and third term of the ELBO are the Kullback-Leibler (KL) divergneces between the variational distribution and true prior for each latent embedding. Following previous work, we use the Gaussian prior with $p(\mathbf{z}_i) = \mathcal{N}(\mathbf{z}_i \mid 0, \mathbf{I})$ and assume the posterior approximation $q(\mathbf{z}_i \mid \mathbf{x}_i, \mathbf{A})$ as Gaussian, which brings the closed form of KL-divergence. It is also worth noting that the inference model for node popularity takes $\mathbf{I}$ instead of $\mathbf{X}$ for its input (see second term in RHS in Equation 6). This is mainly because node popularity is one of the characteristics of node itself, and the dependence on $\mathbf{X}$ has been intentionally dropped.

## 4.3 Training and Inference

Our model follows a 'winner-take-gradient' training strategy [26] by positing the problem to hard EM algorithm for end-to-end learning. We thus introduce additional parameter $\mathbf{e}$ as an indicator, where $\Pr(\mathbf{e}_{ij} = 1) = p_{ij}^h$. At each interaction, $\mathbf{e}_{ij} \in \{0, 1\}$ is sampled from Bernoulli. The overall learning process is provided in Algorithm 1.

---

**Algorithm 1** Training procedure for D-VGAE

---

**Require:** $\mathbf{X} \in \mathbb{R}^{N \times D}, \mathbf{A} \in \mathbb{N}^{N \times N}$
**Ensure:** $\boldsymbol{\phi}^h, \boldsymbol{\phi}^p$ for a single GNCN encoder while normalizations are performed separately for homophily and node popularity.
  Initialize $\boldsymbol{\phi}^h, \boldsymbol{\phi}^p$
  **while** not converged **do**
    Obtain batch of nodes
    **Expectation step:**
    **for** node $i, j$ in a batch **do**
        Sample $\mathbf{e}_{ij}$ from the variational distribution with the latest setting of $p_{ij}^h$.
    **end for**
    **Maximization step:**
    Take average of gradients from the batch to maximize $\mathcal{L}(\boldsymbol{\phi}, \boldsymbol{\theta}; \mathbf{A})$ with $\mathbf{e}_{ij}$ from Expectation step.
  **end while**

---

Specifically, when we sample $\mathbf{e}_{ij}$, we borrow the latest setting of $p^h$ which can be obtained from Equation 3 with temperature added. We anneal the temperature from high value to low value in a way the model can sufficiently explore the two scenarios and avoid *over-confidence* issue. We also reveal how our 'winner-take-gradient' training strategy (hard EM) achieves better performance than soft counterpart through our experimental results in Appendix B.4.

**Table 1: Dataset statistics**

| Dataset | #Nodes | #Edges | #Features | #Classes |
|---|---|---|---|---|
| Cora | 2,708 | 5,429 | 1,433 | 7 |
| CiteSeer | 3,327 | 4,732 | 3,703 | 6 |
| PubMed | 19,717 | 44,338 | 500 | 3 |
| CS | 18,333 | 163,788 | 6,805 | 15 |
| Physics | 34,493 | 495,924 | 8,415 | 5 |
| Computers | 13,752 | 491,722 | 767 | 10 |
| Photo | 7,650 | 238,162 | 745 | 8 |

## 5 EXPERIMENTS

### 5.1 Experimental Setup

*Configuration.* Before presenting our experimental results, we first describe the experimental setup for the empirical evaluations of our proposed method. The model is implemented and trained based on PyTorch Geometric [9] library. The experiments are conducted using an NVIDIA RTX A6000. Every experiment in this paper follows the experimental protocols in [22] with 10% of links for testing, and 5% of links for validation. The models are trained on 85% of links, and the associated node features. In validation and testing, we compare the positive edges against the same number of negative edges which have been sampled randomly from pairs of unconnected nodes. Following a standard manner of learning-based link prediction, we also perform 10 runs of experiments and report *area under the ROC curve* (AUC) and *average precision* (AP) on the test set. The embedding of each node is learned in 256-dimensional latent space. Further details on configuration is provided in the Appendix.

*Datasets.* We evaluate the proposed approach based on benchmark network datasets with node features for undirected link prediction. Table 1 summarizes each dataset used across our experiments. The datasets in top three rows in Table 1 have always been used in previous methodologies for performance evaluations. Cora [33], CiteSeer [11], and PubMed [35] are citation network datasets containing list of citation links between documents and bag-of-words (BOW) feature vectors for each document. The CS, Physics, Computers, Photo datasets have been used in [1] along with Cora, CiteSeer, and PubMed. CS and Physics are co-authorship graphs based on the Microsoft Academic Graph from the KDD Cup 2016 challenge. In CS and Physics datasets, authors are connected if the authors co-published a paper, where paper keywords for each author's papers are aggregated for node features. Computers and Photo are segments of the Amazon co-purchase graph in [32]. In Computers and Photo datasets, nodes represent goods, edges are generated if the two goods are frequently bought together, and bag-of-words encoded product reviews are used as node features. While these four datasets have been originally used for node classifications when first appread in [44], we use these datasets for link prediction as in [1] removing node labels. As in the previous studies, if the original links are directed, they were treated as undirected links. Node labels are only being used in specific downstream tasks: classification and clustering.

### 5.2 Baselines

We compare D-VGAE against the competitive baseline methods including the current SOTA model. GAE and VGAE are introduced in [22], where both GAE and VGAE use the Graph Convolutional Network (GCN) [23] encoder and a simple inner-product decoder. The GCN in GAE/VGAE has been replaced by a simple linear model in LGAE [42]. Experimental results in [42] shows how the simple first-order linear encoders are effective as the popular GCNs. ARGA [37] is an adversarial graph embedding framework for graph data, which enforces latent representation to match a prior distribution. GIC [31] leverages cluster-level node information through differentiable $K$-means. sGraphite [7] tries to enlarge the normal neighborhood through maximizing the mutual information. SIG-VAE [16] combines semi-implicit variational inference (SIVI) [53] and normalizing flow (NF) [20, 38, 41] into the VGAE framework, and also proposes a new Bernoulli-Poisson link decoder. MSVGA [15] tries to learn multiple sets of low-dimensional vectors of different dimensions, which is extended by SPN-MVGAE [51] that incorporates conditional sum-product networks as constraints. GIC+WP is reported as the best performing model in [36], which applies a new pooling scheme called WalkPool on embeddings learnt from GIC. Among these baseline models, GNAE/VGNAE [1] achieves the best performance both on Cora and CiteSeer datasets; GIC+WP[36] achieves the best performance on PubMed. GIC+WP cannot be compared directly to other end-to-end baselines. Models using class labels associated to nodes haven't been included as our baseline methods.

### 5.3 Numerical Results

*Main Results.* We quantitatively evaluate D-VGAE through experiments on the benchmark datasets for link prediction. Our first set of experiments compares D-VGAE against existing baseline methods. The experiment has been standardized by [22], where 10% of links were used for testing and other 5% of links for validation. We perform link predictions using the baseline approaches and D-VGAE using the rest of 85% of the links and the features of the associated nodes. With the observed link and its full feature data used as $\mathbf{A}$ and $\mathbf{X}$ for the input through D-VGAE, we obtain each probability of the component of $\hat{\mathbf{A}}$. When 10% of links were sampled for testing, the same number of non-link were sampled. We use these links and non-links as the ground-truth and compare them to our predictions. Through binary classification, for each round of experiments, AUC (area under the ROC curve) and AP (average precision) can be obtained. We perform 10 rounds of link prediction for each dataset across all the models in this study, and report the overall results in Table 2. The best results in each metric are marked in bold. We set all the hyperparameters the same throughout three benchmark datasets, without further tuning on each dataset. From Tables 2 and 3, we observe D-VGAE achieves state-of-the-art results consistently across all the benchmark datasets.

*Larger Graph.* We further validate D-VGAE on larger datasets. We use CS, Physics, Computers, and Photo dataset for evaluation, where the statistics of each dataset have been provided in Table 1. From Table 2, we found GNAE/VGNAE generally achieves results comparing to other end-to-end models. We thus mainly compare

Table 2: Comparing our proposed method against the benchmark models. The results (except for GNAE/VGNAE) are taken from the respective original papers. All of the models follows the experimental protocols in [22] with 10% of the links for testing, and 5% of the links for validation. Our model is tested in two ways: end-to-end (E2E) and with WalkPool (+WP).

| Model | Cora | | CiteSeer | | PubMed | |
|---|---|---|---|---|---|---|
| | AUC | AP | AUC | AP | AUC | AP |
| GAE [22] NIPS-W'16 | $91.0 \pm 0.02$ | $92.0 \pm 0.03$ | $89.5 \pm 0.04$ | $89.9 \pm 0.05$ | $96.4 \pm 0.00$ | $96.5 \pm 0.00$ |
| VGAE [22] NIPS-W'16 | $91.4 \pm 0.01$ | $92.6 \pm 0.01$ | $90.8 \pm 0.02$ | $92.0 \pm 0.02$ | $94.4 \pm 0.02$ | $94.7 \pm 0.02$ |
| ARGA [37] IJCAI'18 | $92.4 \pm 0.003$ | $93.2 \pm 0.003$ | $91.9 \pm 0.003$ | $93.0 \pm 0.003$ | $96.8 \pm 0.001$ | $97.1 \pm 0.001$ |
| LGAE [42] NIPS-W'19 | $92.05 \pm 0.93$ | $93.32 \pm 0.86$ | $91.50 \pm 1.17$ | $92.99 \pm 0.97$ | $95.88 \pm 0.20$ | $95.89 \pm 0.17$ |
| SIG-VAE [16] NeurIPS'19 | $96.04 \pm 0.04$ | $95.82 \pm 0.06$ | $96.43 \pm 0.02$ | $96.32 \pm 0.02$ | $97.01 \pm 0.07$ | $97.15 \pm 0.04$ |
| sGraphite [7] IJCNN'20 | $93.7 \pm 0.13$ | $93.5 \pm 0.11$ | $94.1 \pm 0.13$ | $95.4 \pm 0.09$ | $94.8 \pm 0.03$ | $96.3 \pm 0.02$ |
| GIC [31] PAKDD'21 | $93.5 \pm 0.6$ | $93.3 \pm 0.7$ | $97.0 \pm 0.5$ | $96.8 \pm 0.5$ | $93.7 \pm 0.3$ | $93.5 \pm 0.3$ |
| GNAE [1] CIKM'21 | $95.38 \pm 0.02$ | $95.91 \pm 0.02$ | $96.81 \pm 0.02$ | $97.19 \pm 0.03$ | $97.27 \pm 0.01$ | $97.21 \pm 0.01$ |
| VGNAE [1] CIKM'21 | $95.32 \pm 0.02$ | $95.36 \pm 0.03$ | $96.96 \pm 0.02$ | $97.01 \pm 0.03$ | $97.24 \pm 0.02$ | $97.11 \pm 0.02$ |
| MSVGA [15] WSDM'22 | $95.3 \pm 0.05$ | $95.4 \pm 0.04$ | $95.4 \pm 0.03$ | $96.1 \pm 0.04$ | - | - |
| GIC+WP [36] ICLR'22 | $95.90 \pm 0.5$ | - | $95.94 \pm 0.53$ | - | $98.72 \pm 0.10$ | $98.72 \pm 0.10$ |
| SPN-MVGAE [51] WWW'23 | $94.6 \pm 0.02$ | $95.6 \pm 0.02$ | $95.5 \pm 0.02$ | $96.3 \pm 0.01$ | $97.0 \pm 0.00$ | $97.5 \pm 0.01$ |
| **D-VGAE (E2E)** | $\mathbf{96.34} \pm 0.03$ | $\mathbf{96.13} \pm 0.03$ | $\mathbf{97.37} \pm 0.03$ | $\mathbf{97.29} \pm 0.03$ | $98.07 \pm 0.04$ | $97.87 \pm 0.05$ |
| **D-VGAE (+WP)** | - | - | - | - | $\mathbf{98.94} \pm 0.04$ | $\mathbf{98.93} \pm 0.05$ |

Table 3: Link prediction results with same protocol.

| Model | CS | | Physics | | Computers | | Photo | |
|---|---|---|---|---|---|---|---|---|
| | AUC | AP | AUC | AP | AUC | AP | AUC | AP |
| VGAE [22] | 95.94 | 95.37 | 96.33 | 95.81 | 92.32 | 92.19 | 94.38 | 93.69 |
| VGNAE [1] | 96.12 | 95.53 | 96.04 | 95.23 | 94.90 | 94.74 | 95.97 | 95.33 |
| **D-VGAE (E2E)** | **97.77** | **97.43** | **97.39** | **96.87** | **97.18** | **96.67** | **97.70** | **97.17** |

D-VGAE against GNAE/VGNAE and vanilla GAE/VGAE. The experiments are conducted following the same experimental protocols in [22]. The hyperparameters were kept the same from the previous set of experiments, without further tuning for each datasets. Table 3 shows how our model outperforms other models. Moreover, the improvement is much more pronounced than that seen with standard datasets. The results show how our model consistently outperforms the baseline with respect to all metrics, which reflects the efficacy of decoupling of node embeddings.

*Node classification .* While our node embeddings have been optimized for predicting links, we can leverage the node embeddings for other downstream tasks such as node classification and clustering. Here, we use all the available link information (100% links for training) for learning the node embeddings, and use the embeddings for node classification. Following [44], we use the node embeddings as input and train a simple classifier with training data which have been randomly selected: 20 samples per class. We compare our results to other downstream classification results reporting the average accuracy over 20 runs[1]. The embeddings learned by D-VGAE exhibit competitive results as shown in Table 4, where we achieve higher performances than previous models except one dataset.

---

[1]Semi-supervised approaches were not compared in this experiments.

Table 4: Node Classification accuracy (in %). The results are directly borrowed from the corresponding papers. Experiments follow the protocol in [44]. 'h only' means we only use angular node embedding (homophily).

| Model | Accuracy | | | | | | |
|---|---|---|---|---|---|---|---|
| | Cora | CiteSeer | PubMed | CS | Physics | Computers | Photo |
| DGI [49] | 80.0 | 70.5 | 76.8 | 88.7 | 91.8 | 77.9 | 86.8 |
| SIG-VAE [16] | 79.7 | 70.4 | 79.3 | - | - | - | - |
| GIC [31] | 80.7 | **70.8** | 77.4 | 89.3 | 92.4 | 79.5 | 89.0 |
| D-VGAE (h only) | **82.6** | 70.2 | **80.7** | **90.2** | **93.8** | 79.6 | **90.1** |

## 5.4 Ablative Analysis

We provide indepth ablative analysis to determine which aspects of our model contribute to the model performance. In the first analysis, we implement the link prediction model inspired by [12] in VGAE framework, which is compared against D-VGAE. In the second analysis, we perform link prediction only using the node embedding from homophily while training in a same manner as in D-VGAE, and provide an answer to RQ2. Finally in the third analysis, we compare the performance of WalkPool [36] by using different preprocessings. Further ablative analysis can be found in the Appendix.

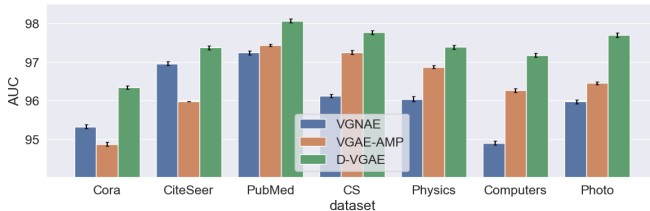

**Figure 3: Link prediction using VGAE-AMP [12] (center bar in orange for each dataset)**

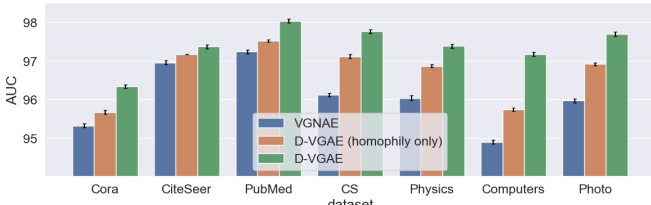

**Figure 4: Link prediction with homophily from D-VGAE (center bar in orange for each dataset)**

Figure 3 compares the performance of three models, where VGAE-AMP is our VGAE implementation of AMP in [12]. In VGAE-AMP, we take into account both homophily and node popularity in a same manner as in AMP, where the logit of link probability is defined by linear sum of node popularity and homophily. In Cora and CiteSeer datasets, we observe VGAE-AMP underperforms VG-NAE [1], while in PubMed, it slightly performs better than VGNAE.

The results reflects how decoupling the two effects contributes to the model performance answering **RQ1**. The other research question to be answered was **RQ2)** whether the embedding for homophily itself becomes more accurate when the node popularity effect gets removed during training. The results in Figure 4 provide the answer, where we observe the link prediction performed only using $z^h$ without node popularity effect have already improve the performance from the previous SOTA. We believe the embedding for homophily itself becomes more accurate when the node popularity effect gets removed during training letting homophily better captured.

*WalkPool.* In Table 2, D-VGAE+WP on PubMed achieved higher performance than the results from [36]. The results in Table 5 show that the results we achieved are not merely due to the GNCN (by comparing against VG-NAE). We also stress that VGAE variants cannot be directly compared with the results from [36]. VGAE and D-VGAE are end-to-end models, while the model in [36] requires node embedding obtained from other models such as GIC, VGAE. The authors reported GIC+WP as their best performing model, which we had as one of our baselines in Table 2.

**Table 5: Applying WalkPool**

| Model | Method | PubMed | |
|---|---|---|---|
| | | AUC | AP |
| GIC [31] | −WP | 93.00 | 92.32 |
| | +WP | 98.72 | 98.72 |
| VGNAE [1] | −WP | 95.32 | 95.36 |
| | +WP | 98.89 | 98.88 |
| D−VGAE (ours) | −WP | 98.07 | 97.87 |
| | +WP | **98.94** | **98.93** |

### 5.5 Qualitative Evaluations

For qualitative study, we use IMDb network data, and learn node embeddings of $z^p$ and $z^h$. This dataset has been crawled in 2018, where the edge means the two actors have appeared in a movie. The dataset has 11,384 nodes and 68,264 undirected edges, which is summarized in Figure 5 with degree. The size and color of node represents degree. The dataset we use in this study is featureless network data, where we take featureless approach replacing input

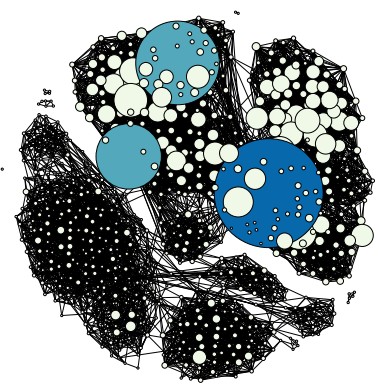

**Figure 5: IMDb actor network**

X with the identity matrix. We want to verify whether the node popularity we learn is beyond observable degree. The top 10 actors in terms of node degree in the data happen to be all Indian actors with Prakash Raj the highest followed by Mohan Joshi. Interestingly, the popular nodes we found based on $z^p$ were quite different from the actors based on degree. The node with highest node popularity was Pierce Brosnan. In top 10, we additionally found Nicolas Cage, Jonny Depp, Sylvester Stallone. We qualitatively show that D-VGAE can capture popular nodes of which the links are established beyond homophily, e.g., across various genres, countries. We also observe that the popularity is not a mere artifact of degree.

## 6 CONCLUSION

In this paper, we discuss the intrinsic limitation of inner product based decoder in VGAE, where the norm and the cosine similarity both try to explain the probability of link. We propose a novel framework of VGAE through decoupling the two effects and associate each component to its own phenomena: node popularity and homophily. The decoupling also avoids unexpected domination effect in message passing. To effectively decouple the two effects, we propose a two-stage generative process accounting for each individually. We perform end-to-end learning using the hard EM algorithm consistently achieving SOTA results on standard datasets. The link prediction which only uses the node embedding from homophily in our model already outperforms previous models. We hope our study can inspire other researchers to perform further studies in VGAE.

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

# A EXPERIMENTAL DETAILS

## A.1 Implementation

We implement D-VGAE based on PyTorch Geometric [9] 2.0.4. In our experiments with D-VGAE, we use Graph Normalized Convolutional Network (GNCN) as our encoder which is introduced in [1]. In our experiments with D-VGAE, we use instances of `GNCNEncoder`. We define a class, `DVGAE` that takes an encoder and a decoder of our own following the Equation 3 and 4. The decoder uses Gumbel-trick to predict edge under homophily or node popularity. To select one property, our generative model performs hard sampling. Following the previous approaches [3, 17, 25], the temperature $\tau$ is annealed from a high temperature to small temperature.

## A.2 Additional Experimental Details

The experiments are conducted with PyTorch 1.10.2 using an NVIDIA RTX A6000. The latent embedding dimension is fixed to 256 in our experiments. The model uses validation set for early stopping. For the baseline models, we also tested with different hyperparameters, but found the hyperparameters from the original papers perform the best. The Gumbel-Softmax temperature has been annealed from {10.0, 5.0, 2.0, 1.0} to {2.0, 1.0, 0.5, 0.1, 0.01}. We found having high temperature as 2.0 and low temperature as 0.5 generally achieves competitive performance. We anneal the temperature to 0.01 when reporting our results.

# B EXTENDED EXPERIMENTS

## B.1 VGNAE with High-Degree Nodes

When the similarity between nodes are mearsured using the normalized innder product, the link probability is bounded in a range for every nodes regardless of the node degree. While it brings performance improvement for low-degree nodes, it causes performance degradation for high-degree nodes. In this experiments, we perform link predictions with links associated with high-degree nodes. For

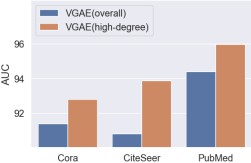 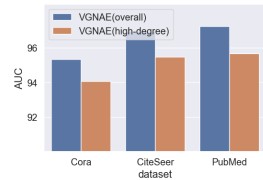

(a) In VGAE, we observe higher performance on links with high-degree nodes

(b) In VGNAE, we observe lower performance on links with high-degree nodes

**Figure 6: For two models, we compare the overall performance and the performance on links associated with high degree nodes. While VGAE achieve performance improvement, VGNAE shows performance degradation.**

each of dataset, we use node with degree above 5 as high-degree nodes.

As shown in Figure 6a, VGAE achieves higher performance when the links are associated with high-degree nodes. This is expected as the magnitude of high-degree nodes can account for the link probability, which have been shown in Figure 1a. However, we observe the opposite behavior when we perform the same testing using VGNAE with normalized inner product. As shown in Figure 6b, the performance for high-degree nodes are always worse than overall performance. In fact, in certain cases, such as for the case with PubMed, performance on low-degree nodes (or isolated nodes) are better than the high-degree nodes.

## B.2 Node Popularity : snp($\cdot$)

In Equation 4, we have snp($\cdot$) which reflects the node popularity. We try three functions for snp($\cdot$). As the link of our interest is undirected, the probability of link under node popularity is affected by one of the two nodes. In this regard, we try adding the two $z^p$s for snp($\cdot$). Another way of defining the node popularity function is to have sum of square values of $z^p$s. In this approach, bias should be added in the sigmoid function to ensure the value can be between 0 and 1. The other function we try is obtaining $z^p$s using vanilla-encoder without normalization. In this approach, we observe performance improvement on CiteSeer dataset (see Table 6). For CiteSeer dataset, the normalization constraint restrict the model to find best $z^p$ that reconstruct the links.

**Table 6: `D-VGAE (no norm)` excludes nomalization for node popularity.**

| Model | Cora | | CiteSeer | | PubMed | |
|---|---|---|---|---|---|---|
| | AUC | AP | AUC | AP | AUC | AP |
| VGNAE | 95.32 ± 0.02 | 95.36 ± 0.03 | 96.96 ± 0.02 | 97.01 ± 0.03 | 97.24 ± 0.02 | 97.11 ± 0.02 |
| D-VGAE | **96.34** ± 0.03 | **96.17** ± 0.03 | 97.37 ± 0.03 | 97.29 ± 0.03 | **98.07** ± 0.04 | **97.87** ± 0.05 |
| D-VGAE (no norm) | 95.55 ± 0.04 | 95.91 ± 0.04 | **97.40** ± 0.03 | **97.45** ± 0.03 | 97.63 ± 0.05 | 97.65 ± 0.05 |

## B.3 Message Passing the Node Popularity

In this ablative study, we perform link prediction using a model based on our model, D-VGAE, where we apply propagation scheme both on homophily and node popularity. We denote this variant as

D-VGAE (PNP) in Table 7 and compare with D-VGAE which only performs propagation on embeddings for homophily.

**Table 7: `D-VGAE (PNP)` applies message passing scheme on node popularity, which we compare to our proposed model with propagation scheme only applied on homophily.**

| Model | Cora | | CiteSeer | | PubMed | |
|---|---|---|---|---|---|---|
| | AUC | AP | AUC | AP | AUC | AP |
| VGNAE | $95.32 \pm 0.02$ | $95.36 \pm 0.03$ | $96.96 \pm 0.02$ | $97.01 \pm 0.03$ | $97.24 \pm 0.02$ | $97.11 \pm 0.02$ |
| D-VGAE | $\mathbf{96.34} \pm 0.03$ | $\mathbf{96.17} \pm 0.03$ | $\mathbf{97.37} \pm 0.03$ | $\mathbf{97.29} \pm 0.03$ | $\mathbf{98.07} \pm 0.04$ | $\mathbf{97.87} \pm 0.05$ |
| D-VGAE (PNP) | $95.47 \pm 0.02$ | $95.53 \pm 0.02$ | $96.59 \pm 0.04$ | $96.89 \pm 0.03$ | $97.82 \pm 0.05$ | $\mathbf{97.72} \pm 0.05$ |

As shown in Table 7, we observe that D-VGAE always outperforms other methods including our extension. We observe D-VGAE (PNP) performs bettern than VGNAE in two datasets: Cora and PubMed. The results support our hypothesis that the node popularity is the property of the node itself, which is not affected by their neighbors.

## B.4 Stochastic Sampling

Our model follows a 'winner-take-gradient' training strategy [26] by positing the problem to hard EM algorithm. In our E-like step, we perform sampling for $\mathbf{e}_{ij}$ for a given pair of node $i$ and $j$, where we use Gumbel-trick for hard sampling. With the indicator vector $\mathbf{e}_{ij}$, we let the gradient flow only through the selected property (homophily vs node popularity). Here, we verify how our approach is effective by comparing D-VGAE against the variant which uses probabilistic values instead of the $\mathbf{e}_{ij}$.

**Table 8: `D-VGAE (softmax)` uses softmax for inferring the property between homophily and node popularity.**

| Model | Cora | | CiteSeer | | PubMed | |
|---|---|---|---|---|---|---|
| | AUC | AP | AUC | AP | AUC | AP |
| VGNAE | $95.32 \pm 0.02$ | $95.36 \pm 0.03$ | $96.96 \pm 0.02$ | $97.01 \pm 0.03$ | $97.24 \pm 0.02$ | $97.11 \pm 0.02$ |
| D-VGAE | $\mathbf{96.34} \pm 0.03$ | $\mathbf{96.17} \pm 0.03$ | $\mathbf{97.37} \pm 0.03$ | $\mathbf{97.29} \pm 0.03$ | $\mathbf{98.07} \pm 0.04$ | $\mathbf{97.87} \pm 0.05$ |
| D-VGAE (softmax) | $94.93 \pm 0.04$ | $94.55 \pm 0.03$ | $96.92 \pm 0.03$ | $96.82 \pm 0.03$ | $97.48 \pm 0.05$ | $97.36 \pm 0.04$ |

D-VGAE (softmax) performs worse than VGNAE on Cora dataset. We also observe D-VGAE (softmax) slightly underperforms than VGNAE on CiteSeer. However, on PubMed dataset, D-VGAE performs better than VGNAE. On evey dataset, our proposed model D-VGAE with Gumbel-trick sampling always performs better than D-VGAE (softmax). We believe this is mainly due to the decoupling effect, which clearly devides the two property through sampling, and our generative process.

## B.5 Performing Normalization in GNCN

In this ablative study, we perform inference and obtain full node embeddings, and split the embeddings into homophily embedding and node popularity embedding. This is quite different from our proposed approach, where we perform GNCN for homophily and different GNCN for node popularity. We compare the performance from the two approaches.

The results in Table 9 provides the results when the embedding is normalized together first and split later. While this approach also takes into account the homophily and node popularity , and

**Table 9: `D-VGAE (norm and split)` performs GNCN and split the embedding vector into two: homophily and node popularity**

| Model | Cora | | CiteSeer | | PubMed | |
|---|---|---|---|---|---|---|
| | AUC | AP | AUC | AP | AUC | AP |
| VGNAE | $95.32 \pm 0.02$ | $95.36 \pm 0.03$ | $96.96 \pm 0.02$ | $97.01 \pm 0.03$ | $97.24 \pm 0.02$ | $97.11 \pm 0.02$ |
| D-VGAE | $96.34 \pm 0.03$ | $96.17 \pm 0.03$ | $97.37 \pm 0.03$ | $97.29 \pm 0.03$ | $98.07 \pm 0.04$ | $97.87 \pm 0.05$ |
| D-VGAE (norm and split) | $95.13 \pm 0.05$ | $94.42 \pm 0.04$ | $96.82 \pm 0.03$ | $96.21 \pm 0.04$ | $97.23 \pm 0.05$ | $96.87 \pm 0.05$ |

looks similar, major difference lies in the encoder. The embedding of homophily and node popularity should be learned separately and each should have its own normalization.

## B.6 Clustering Visualization

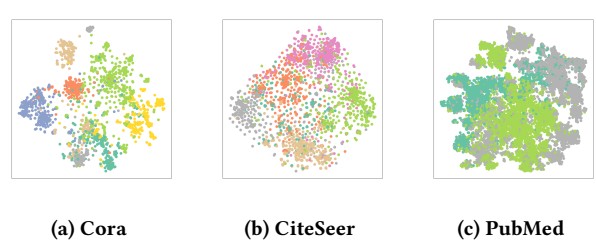

| (a) Cora | (b) CiteSeer | (c) PubMed |

**Figure 7: Angular node embeddings ($z^h$) from D-VGAE are visualized using 2D $t$-SNE, where we compare with the groundtrugh labels in different colors.**

We perform 2D $t$-SNE projections of the node embeddings obtained through D-VGAE. For visualization, we only use the angular node embedding ($z^h$), and no extra process has been applied. When compared with the ground-truth label, we qualitatively verify how D-VGAE performs on clustering.

