# OpenReview forum: "Decouped Variational Graph Autoencoder for Link Prediction"
_ACM.org/TheWebConf/2024/Conference — TheWebConf24 Oral_

### Official Review · Reviewer_3zsv · 2023-11-21

**Novelty:** 5
**Technical Quality:** 5

**Review:**

This paper introduces a novel approach within the Variational Graph Autoencoder (VGAE) framework to improve link prediction in graph-structured data. The key innovation is the decoupling of norm and angle in node embeddings, addressing a fundamental limitation in existing VGAE models where the cosine similarity and norm compete in explaining link probability. The authors propose a hard expectation maximization learning method to distinguish between the effects of node homophily and popularity on link formation. Extensive experiments on real-world datasets demonstrate that the proposed model outperforms existing state-of-the-art methods in link prediction, while also achieving comparable results in other tasks like node classification and clustering.


Strengths
1. The observation in Figure 1, where the norm of node embedding obtained via VGAE increases with its degree, is a compelling finding.
2. Introducing the concept of decoupling norm and angle in node embeddings under the VGAE model is an innovative approach. The methodology employed in this study shows considerable promise.
3. Experimental results demonstrate that the proposed method outperforms baseline models, showing its effectiveness in link prediction tasks.


Weaknesses
1. The definition 1 for node popularity in the graph is unclear and informal.
2. Regarding the insights derived from Figure 1, integrating degree information to refine node embedding learning, particularly in the context of node popularity, would be advantageous.
3. The paper's approach is heavily reliant on the assumptions about node popularity and homophily. A more comprehensive theoretical analysis would substantiate these assumptions and strengthen the paper's arguments.

**Questions:**

see above

**Reviewer Confidence:**

3: The reviewer is confident but not certain that the evaluation is correct

**Scope:**

4: The work is relevant to the Web and to the track, and is of broad interest to the community

---

### Official Review · Reviewer_pJUm · 2023-11-26

**Novelty:** 4
**Technical Quality:** 4

**Review:**

This paper discusses the limitation of inner product based decoder in VGAE, and introduces a generative algorithm to decouple node popularity and homophily through a two-stage generation process. Empirical results show that the proposed D-VGAE outperforms SOTA methods on link prediction and achieves comparable performances on node classification and clustering.

Strength:
S1. The paper observes of the limitation of VGAE, especially in representation learning for high-degree and low-degree nodes.

S2.  The proposed two-step generative process for homophily and node popularity is sound.

Weakness:
W1. In light of the studied problem, author should demonstrate the effectiveness of the proposed approach in imbalanced datasets and empirically show its advantage in high/low-degree node representation compared to SOTA baselines. However, authors only demonstrate the experimental results on common benchmarks. The improvement is limited on these datasets.

W2. This paper is hard to follow, especially in describing the generation process of the proposed framework and the decoupling effects.

W3. In-depth analyses in some experiments are missing. For example, in Section 5.4, why AMP and WalkPool are chosen for comparison? What differences do they bring compared to the proposed D-VGAE?

**Questions:**

Please refer to the weak points above.

**Reviewer Confidence:**

3: The reviewer is confident but not certain that the evaluation is correct

**Scope:**

3: The work is somewhat relevant to the Web and to the track, and is of narrow interest to a sub-community

---

### Official Review · Reviewer_9uAc · 2023-11-27

**Novelty:** 5
**Technical Quality:** 6

**Review:**

This paper proposes a VGAE-based method named D-VGAE to make link prediction more effective. The method decouples the inner product within VGAE into two components: the cosine similarity and norm. Using a similarity-based decoder to learn embedding space for homophily and a norm-based decoder to learn embedding for node popularity. Then a two-stage generative process is proposed for link prediction. A large number of experimental results show that the proposed method not only is better than all existing methods in link prediction but also has a comparable performance on other downstream tasks.

Strengths
1. This paper is the first one to incorporate node popularity in VGAE. By decoupling the inner product in the VGAE model into node popularity and homophily, the method achieves better results than the current SOTA model.
2. This paper provides a comprehensive and reasonable explanation of the proposed method. For example, from lines 441 to 456, the reason for the use of the two-component vector z𝑝 is analyzed in detail.
3. For all aspects of the method, the effectiveness has been demonstrated through experiments. For example, in Appendix B.2. , the choice of the snp function is determined experimentally.

Weaknesses:
1. The analysis of the experimental results was not deep enough. For example, from the experimental results of Table 2, the paper only illustrates the superiority of the proposed method but does not provide a more detailed analysis of why other methods perform poorly.
2. Figure 2 is not described in more detail. It is suggested a detailed description of the individual graphical representations in the diagram.
3. It is suggested a more nuanced description of node popularity in line 330.

**Questions:**

Please refer to the strengths and the weaknesses.

**Reviewer Confidence:**

3: The reviewer is confident but not certain that the evaluation is correct

**Scope:**

4: The work is relevant to the Web and to the track, and is of broad interest to the community

---

### Official Review · Reviewer_nCkT · 2023-11-27

**Novelty:** 3
**Technical Quality:** 3

**Review:**

This article posits that the VGAE method overlooks the impact of node attribute norms on link prediction. The authors argue that nodes with smaller degrees tend to have smaller norms, while nodes with larger degrees tend to have larger norms. Larger norms disproportionately dominate the node representation learning process, diminishing the influence of nodes with smaller norms. Based on this observation, the authors propose the D-VGAE method, decomposing the inner product in VGAE into cosine similarity and norm. They use cosine similarity to measure node similarity and norm to quantify the node popularity. The aim is to decouple link prediction into these two aspects, predicting them separately to enhance link prediction accuracy.

However, the actual situation does not necessarily align with the observations made by the authors. Taking Figure 1(a) as an example, the relationship between node norms and their degrees does not perfectly exhibit the viewpoint expressed by the authors. In the case of Cora, the relationship exhibits a clear V-shape, so node norms may not necessarily be smaller when the node degree is small. For PubMed, when the node degree ranges from 0 to 25, there are numerous nodes with equal norms, contradicting the authors' viewpoint. Therefore, the necessity of decoupling the link prediction process remains open to further discussion.

In summary, the pros and cons of this article are as follows:

**Pros:**
1) Comprehensive experiments.
2) Displays some level of originality.

**Cons:**
1) Spelling errors in the title.
2) Writing lacks clarity, with some content not adequately explained.
3) Significance of this paper is needed to be improved.

I have read the response. I keep my original score.

**Questions:**

1) How to get the node latent variables for node popularity $z_i^p$, how to get the approximate distribution $q_\phi(Z^p|I,A)$?
2) What is the exact meaning of node popularity in graph? Would you please explain the meaning of the Definition 1 from line 333 to line 336?
3) Would you explain that in Table 2, why D-VGAE(+WP) has no results on Cora and CiteSeer?

**Reviewer Confidence:**

4: The reviewer is certain that the evaluation is correct and very familiar with the relevant literature

**Scope:**

3: The work is somewhat relevant to the Web and to the track, and is of narrow interest to a sub-community

---

### Official Review · Reviewer_B4Em · 2023-11-29

**Novelty:** 6
**Technical Quality:** 6

**Review:**

Summary

The paper addresses the challenge of link prediction in graph-structured data, focusing on the limitations of the Variational Graph Autoencoder (VGAE). The authors identify a key issue in VGAE's inner-product decoder, which struggles with link prediction for nodes with very high or very low degrees. They observe that for such nodes, the norm and cosine similarity components of the inner-product decoder compete, leading to suboptimal performance. To address this, the paper proposes a generative algorithm within the VGAE framework, decoupling the norm (related to node popularity) and cosine similarity (related to homophily) components. This approach involves using two different embedding spaces for these components, aiming to improve link prediction accuracy. Additionally, the paper introduces a hard Expectation-Maximization (EM) algorithm for end-to-end learning, claiming state-of-the-art results in link prediction on attributed networks.

Pros

(1) Interesting Motivation: The paper tackles a significant and complex issue in graph-structured data analysis, focusing on a real-world challenge in link prediction.

(2) Innovative Solution: The approach of decoupling norm and cosine similarity in the VGAE framework is novel and shows creativity in addressing the identified problem.

(3) Extensive Experiments: The paper reportedly conducts thorough experiments to validate its findings, which is crucial for establishing the efficacy of the proposed solution.

Cons

(1) Limited Application Scenarios: The focus is primarily on VGAE, which might limit the applicability of the findings to other graph-structured data scenarios or models.

(2) Unsupported Assumptions: The paper assumes a direct correspondence between the norm and node popularity, and cosine similarity and homophily, without providing substantial evidence or validation for this assumption. This lack of empirical support could undermine the theoretical foundation of the proposed solution.

**Questions:**

(1) this paper mentions that VGAEs exhibit underperformance in link prediction on low-degree and high-degree nodes. Then, this paper hypothesizes that this is due to the intrinsic limitation of inner-product decoder in VGAE. Is there any more explanation or support behind that hypothesis?

**Ethics Review Description:**

no issue

**Reviewer Confidence:**

3: The reviewer is confident but not certain that the evaluation is correct

**Scope:**

4: The work is relevant to the Web and to the track, and is of broad interest to the community

---

### Decision · Program_Chairs · 2024-01-22

**Decision:**

Accept (Oral)

**Comment:**

The paper introduces a new method to improve link prediction in graphs by tackling problems in the Variational Graph Autoencoder (VGAE). The authors suggest separating two elements in node embeddings - norm and angle - to overcome issues with VGAE's decoder, especially for nodes with very high or very low degrees. They use this separation to handle two different aspects of graphs: cosine similarity for node similarity and norm for how popular a node is, using a new algorithm in the VGAE framework. The paper claims to achieve the best results in link prediction based on thorough testing on various datasets.

 Reviewers praise the paper for addressing a significant problem in a creative and new way. They appreciate the detailed experiments that support the paper's claims. The idea of separating norm and cosine similarity is seen as an original and potentially important advancement in the field.

 However, the reviewers have some concerns. One points out that focusing mainly on VGAE could limit how the findings apply to other graph data models. There are also doubts about some assumptions the paper makes, like the direct link between norm and node popularity, and cosine similarity and similarity in node characteristics, which aren't backed up with enough evidence. This might weaken the paper's theoretical basis.

 Other issues raised include not analyzing experimental results deeply enough, unclear writing, and a need for better explanations of key ideas like node popularity. The authors agree with these points and plan to improve the manuscript accordingly, with clearer explanations and a more detailed description of their model.

 In summary, this paper makes an important contribution to graph data analysis, specifically in the area of link prediction. Its new approach to address VGAE's limitations by separating norm and angle in node embeddings is notable. The paper is original and shows promise, but it needs better theoretical support, clearer writing, and more thorough analysis. The authors are open to making these improvements in their next version.